# CO Adsorption Performance of CuCl/Activated Carbon by Simultaneous Reduction–Dispersion of Mixed Cu(II) Salts

**DOI:** 10.3390/ma12101605

**Published:** 2019-05-16

**Authors:** Cailong Xue, Wenming Hao, Wenping Cheng, Jinghong Ma, Ruifeng Li

**Affiliations:** College of Chemistry and Chemical Engineering, Taiyuan University of Technology, Taiyuan 030024, China; xuecailong35@163.com (C.X.); haowenming@tyut.edu.cn (W.H.); chengwenping@tyut.edu.cn (W.C.); rfli@tyut.edu.cn (R.L.)

**Keywords:** CuCl/AC adsorbent, CO adsorption, monolayer dispersion, isosteric heat, adsorption isotherms

## Abstract

CO is a toxic gas discharged as a byproduct in tail gases from different industrial flue gases, which needs to be taken care of urgently. In this study, a CuCl/AC adsorbent was made by a facile route of physically mixing CuCl_2_ and Cu(HCOO)_2_ powder with activated carbon (AC), followed by heating at 533 K under vacuum. The samples were characterized by X-ray powder diffraction (XRD), inductively coupled plasma optical emission spectrometry (ICP-OES), N_2_ adsorption/desorption, and scanning electron microscopy (SEM). It was shown that Cu(II) can be completely reduced to Cu(I), and the monolayer dispersion threshold of CuCl on AC support is 4 mmol·g^−1^ AC. The adsorption isotherms of CO, CO_2_, CH_4_, and N_2_ on CuCl/AC adsorbents were measured by the volumetric method, and the CO/CO_2_, CO/CH_4_, and CO/N_2_ selectivities of the adsorbents were predicted using ideal adsorbed solution theory (IAST). The obtained adsorbent displayed a high CO adsorption capacity, high CO/N_2_, CO/CH_4_, and CO/CO_2_ selectivities, excellent ad/desorption cycle performance, rapid adsorption rate, and appropriate isosteric heat of adsorption, which made it a promising adsorbent for CO separation and purification.

## 1. Introduction

With the rapid development of C−1 chemistry recently in the chemical industry, carbon monoxide (CO) as a significant resource has been widely applied to prepare a large variety of chemical products, such as formic acid, acetic acid, oxalic acid ester, carbonic acid two methyl ester, anhydride, etc. [1,2]. Most of these preparations need high-purity CO. The main methods of producing CO are the steam reforming of natural gas and coal gasification [3]. In addition, a significant amount of CO is discharged as a byproduct in tail gases from different industrial flue gases including carbon black tail gas, silicon carbide furnace gas, yellow phosphorus tail gas, coke oven gas, blast furnace gas, etc. [4,5,6]. From both processes, the obtained CO is mixed with N_2_, H_2_, CH_4_, CO_2_, and vapor. CO is toxic to humans because it combines with hemoglobin in the blood to form carboxy-hemoglobin hindering the transportation and release of oxygen in the blood, which leads to death [7]. Moreover, even a trace amount of CO can poison the noble catalysts, such as the proton-exchange membrane fuel cells, which restrict the CO content below 0.2 ppm to protect the platinum electrocatalyst [8,9]. Thus, the separation and purification of CO from different gas mixtures have significance both industrially and environmentally.

Among the proven technologies for CO separation and purification, adsorption processes, such as pressure swing adsorption (PSA) and temperature swing adsorption (TSA) have the advantages of convenient operation, low energy consumption, low operating cost, etc., and have been widely used in CO separation [10,11,12,13,14]. Adsorbent plays a crucial role in the adsorption based gas separation process, it has been found that many porous materials, such as activated carbons [15,16], zeolites [17,18], and metal-organic frameworks (MOFs) [19,20], have adsorption capacities to a certain extent. However, it is difficult to separate and purify CO from gas mixtures by using these materials directly since their adsorption capacity and selectivities are low. Cu(I) adsorbents for CO separation have received extensive attention for their high CO adsorption capacity and high selectivity, since CO molecules can form a π-complexation bond with Cu(I) ions on the adsorbent, which are stronger than the interaction caused by van der Waals forces [21,22,23,24,25,26]. More importantly, π-complexation bonds are still weak enough to be broken by normal engineering operations, such as increasing temperature or reducing pressure, and the adsorbed CO can be easily desorbed, which makes it a suitable adsorbent in PSA and TSA systems [23]. Two approaches are used for making Cu(I) adsorbents. In the first process, Cu(I) adsorbents are prepared by impregnating Cu(II) salts into a porous support including zeolites, activated carbons (ACs) and MOFs, etc. [20,25,27], and then reducing Cu(II) to Cu(I) using reducing gases, such as H_2_ or CO. However, it is difficult to control the reduction degree, and Cu(II) is easily over reduced to Cu. In the second process, Cu(I) adsorbents are prepared by direct dispersion and impregnation of CuCl. Hirai et al. [28,29] and Tamon et al. [30] obtained Cu(I)/AC adsorbents by using dispersing reagents, such as concentrated hydrochloric acid or organic solvents, to disperse CuCl onto the AC surfaces, and then drying at 403 K in N_2_. Xie et al. [31] prepared CuCl/zeolite adsorbents by dispersing CuCl powder spontaneously onto the surfaces of zeolites at 623 K in an inert atmosphere, which displayed high adsorption capacity and selectivity for CO. When using CuCl as a starting material, the adsorbent preparation has to be carefully performed in a dry inert atmosphere, to prohibit the oxidation and hydrolysis of Cu(I). In our previous work, we successfully obtained Cu(Ι) π-complexation adsorbents with an aqueous solution of equimolar CuCl_2_ and Cu(HCOO)_2_ as starting materials by the traditional impregnation method followed by activating at the temperature of 583 K [32].

Herein, the purpose of this work is to develop CO adsorbent using a solid-state auto reduction–dispersion method with CuCl_2_ and Cu(HCOO)_2_ as the initial material. Then, X-ray powder diffraction (XRD), inductively coupled plasma optical emission spectrometry (ICP-OES), N_2_ adsorption/desorption and scanning electron microscopy (SEM) were employed to characterize the samples. Pure component CO, CO_2_, N_2_, and CH_4_ adsorption isotherms on the adsorbents were measured in a volumetric method. The CO/CO_2_, CO/CH_4_, and CO/N_2_ selectivities of the adsorbents were predicted by using ideal adsorbed solution theory (IAST). The adsorption isotherms were fitted with the Langmuir–Freundlich model, and the corresponding heats of adsorption were calculated. The cyclic CO adsorption on adsorbent was performed to evaluate its repeated availability during the adsorption and desorption cycles. Furthermore, the CO adsorption rate on adsorbent was discussed and reported.

## 2. Materials and Methods

### 2.1. Materials

Copper formate tetrahydrate (Cu(HCOO)_2_·4H_2_O, 98%) and cupric chloride dihydrate (CuCl_2_·2H_2_O, 99%) were purchased from Alfa Aesar Chemical Co. Ltd.(Ward Hill, MA, USA). Activated carbon (AC) was purchased from Chengde Jingda Activated Carbon Manufacturing Co. Ltd. (Chengde, China).

### 2.2. Preparation of CuCl/AC Adsorbents

CuCl/AC adsorbents were synthesized following two steps. First, the AC was physically mixed with CuCl_2_ and Cu(HCOO)_2_ powder to obtain CuCl/AC adsorbent precursors. Then, the obtained precursors were dried at 373 K and activated in a tube furnace at 533 K for 4 h under vacuum. The obtained precursors and CuCl/AC adsorbents were marked as Cu(II)-x/AC and Cu(I)-x/AC (x = 2, 3, 4, 5, 6), in which the loading of copper is 2, 3, 4, 5, and 6 mmol·g^−1^ AC, respectively. The as-synthesized CuCl/AC adsorbents were stored in vacuum dry storage in a desiccator.

### 2.3. Adsorbent Characterization

Powder X-ray diffraction (XRD) patterns of the samples were recorded by a Shimadzu LabX XRD-6000 system (Kyoto, Japan) in the 2θ range of 5 to 35° using CuKα1 (λ=1.54056 Å) radiation operated at 40 kV and 30 mA. The pore volume and surface area of the samples were calculated from N_2_ adsorption/desorption isotherms measured on a surface area and pore size analyzer (QUADRASORB SI, Quantachrome Inc., Boynton Beach, FL, USA) after activating the samples at 393 K for 4 h under vacuum. The specific surface areas (S_BET_) were determined using the BET (Brunauer–Emmett–Teller) method under relative pressure in the range of 0.01 to 0.20. The adsorbed amount of N_2_ at p/p_0_ = 0.98 was employed to calculate the total pore volume (V_Total_). Scanning electron microscope (SEM, Hitachi S4800, Hitachi Ltd., Tokyo Japan) was used to observe the samples’ morphology. Cu contents were measured by inductively coupled plasma optical emission spectrometry (ICP-OES, Thermo-ICAP6300, Thermo Fisher Scientific Co., Ltd., Waltham, MA, USA).

### 2.4. Adsorption Measurements

Before the adsorption measurements, the samples were degassed under vacuum while heating up to 393 K for 4 h. The CO, CO_2_, CH_4_, and N_2_ adsorption isotherms were measured at the temperature required using a static volumetric apparatus (NOVE1000e, Quantachrome Inc., Boynton Beach, FL, USA). During the adsorption measurements, the temperature was maintained by circulating ethanediol-water from a bath with setting temperature. The adsorption capacity was determined from the adsorption isotherm measured at 298 K. Ultrahigh purity grade CO (99.99%), CO_2_ (99.999%), CH_4_ (99.99%), and N_2_ (99.999%) were used without any purification.

## 3. Results and Discussion

### 3.1. Characterization of Samples

The XRD patterns of the copper loaded AC samples before and after activation are presented in Figure 1. Before activation, the diffraction peaks of Cu(HCOO)_2_ and CuCl_2_ [33,34] can be observed in the Cu(II)-x/AC samples, and the reflection intensities increased with the increase of copper loading. After activation, the diffraction peaks of CuCl_2_ and Cu(HCOO)_2_ disappeared, and the Cu(I)-5/AC sample displayed only a relatively weak peak (2θ = 28.5°) of CuCl [35], suggesting that Cu(HCOO)_2_ and CuCl_2_ were transformed into CuCl after activation. Meanwhile, the absence of CuCl diffraction peak on Cu(I)-2/AC, Cu(I)-3/AC, and Cu(I)-4/AC samples might be due to the well dispersion of CuCl on the AC surface beyond the detection limit of XRD [36]. By further increasing the copper loading to 5 and 6 mmol·g^−1^, the appearance of the characteristic peak of CuCl implies that the crystal size of CuCl on the AC surface increased with the increase of CuCl loading, which was able to be detected by XRD with the CuCl loading higher than 5 mmol·g^−1^.

Figure 2a,b show the representative SEM images of the copper loaded AC samples before and after activation. It can be clearly observed that the particles of copper species present on the AC surface for the Cu(II)-4/AC. However, the particles on the AC surface disappeared after activation, which implies that the activation process contributes to the CuCl dispersion on the AC surface. Figure 2c,d show selected-area and the element mapping analyses of Cu(I)-4/AC. It revealed that the copper particles are uniformly dispersed on the AC surface. The good dispersion of CuCl on Cu(I)-4/AC observed by SEM agreed well with the XRD results in Figure 1.

Figure 3 shows the N_2_ adsorption/desorption isotherms of the Cu(I)-x/AC samples at 77 K. The N_2_ adsorption gradually decreased with increasing CuCl loading. Table 1 lists the textural parameters of AC and Cu(I)-x/AC. It can be observed that the total pore volume (V_Total_) and BET surface area (S_BET_) gradually decreased with the increase of CuCl loading, indicating that CuCl had been loaded into the pores of the parent AC. As the CuCl loading increased, more and more surface within the pores were occupied by CuCl, which may result in a further decrease of V_Total_ and S_BET_. As CuCl was well dispersed on the surface of AC when the CuCl loading was below 4 mmol/g, the average pore sizes decreased with the increase of CuCl loading. Smaller pores were filled with the further increase of the CuCl loading. Therefore, the increased average pore size resulted from the percentage increase of the available larger pores in AC, which is similar to the observation by Ramli et al. [37].

It can be observed from Figure 4 that the CO adsorption capacity of CuCl/AC increased with CuCl loading in the range of 0 to 4 mmol·g^−1^. The maximum value of adsorption capacity was 45.4 cm^3^·g^−1^. With the continuous increase of the CuCl loading, the CO adsorption capacity of CuCl/AC with the copper loading of 5 mmol·g^−1^ was almost the same as that with 4 mmol·g^−1^. The decrease of CO adsorption occurred with further increasing the copper loading to 6 mmol·g^−1^. This phenomenon can be ascribed to the following reason. The more copper loaded, the more active sites of CuCl/AC adsorbents present, which would enhance CO adsorption. However, the increase of copper loading also resulted in a decrease of surface area for CuCl/AC adsorbents, as shown in Table 1. As a result, the increase of the adsorbed amount from the increased active sites and the decrease of adsorbed amount from the decrease of surface area were in a dynamic balance in the copper loading range of 4 to 5 mmol·g^−1^. When the copper loading reached 6 mmol·g^−1^, on the one hand, the amount of adsorbed CO decreased because the decrease of surface area; on the other hand, the Cu(I) started to agglomerate on AC surface with considerable copper loading, resulting in the low utilization of active sites.

In addition, Table 1 lists the measured Cu contents with ICP-OES, which are closely approximate to the values in the raw material. The utilization coefficient of surface CuCl is described from the equation:$$\eta =\frac{q_{CO}}{n_{Cu\mathrm{Cl}}}\times 100\%$$
where η is the utilization coefficient of CuCl, $q_{CO}$ is the actual CO adsorption capacity at 298 K and 100 kPa, $n_{Cu\mathrm{Cl}}$ is the mole of CuCl per gram CuCl/AC adsorbent. According to this equation, the utilization coefficients were calculated and presented in Table 1. It can be seen that the utilization coefficient decreased with the increasing of CuCl loading, which means that the high CuCl loading on AC could not guarantee high utilization of CuCl, since not all Cu(I) species can be utilized.

### 3.2. Adsorption Selectivities of CO to CO_2_, CH_4_, and N_2_

Figure 5a gives the adsorption isotherms of pure CO, CO_2_, CH_4_, and N_2_ on Cu(I)-4/AC in the pressure range of 0 to 100 kPa. The adsorption of CO_2_, CH_4_, and N_2_ on Cu(I)-4/AC increased almost linearly with pressure, while the adsorption isotherm of CO on Cu(I)-4/AC presented a type-Ι isotherm [38], that is the CO adsorption increased sharply with pressure at a low pressure range, implying the adsorption of relatively strong CO-Cu(I) π-complexation, which is propitious to separate CO from CO/CO_2_/CH_4_/N_2_ mixed gas.

The Langmuir–Freundlich (L-F) model and IAST were employed together to calculate the CO/CO_2_, CO/CH_4_, and CO/N_2_ selectivities with the equimolar CO/CO_2_, CO/CH_4_, and CO/N_2_ mixture. The L-F model can be expressed as
$$q=q_{m}\frac{bp^{1/n}}{1+bp^{1/n}}$$
where q is the adsorbed amount, p is the pressure and $q_{m}$ is the saturation adsorbed amount, b is the adsorption affinity and n is the corresponding deviation from the Langmuir isotherm.

First, the adsorption isotherm of pure CO, CO_2_, CH_4_, and N_2_ were fitted by the L-F model [38]. After that, the CO/CO_2_, CO/CH_4_, and CO/N_2_ selectivities were predicted by IAST theory [39,40]. Finally, the relevant selectivities curves along with the increase of pressure were obtained. Figure 5b shows that the CO/CO_2_, CO/CH_4_, and CO/N_2_ selectivities decrease gradually with increasing pressure. Nevertheless, the CO/CO_2_, CO/CH_4_, and CO/N_2_ selectivities on Cu(I)-4/AC were still up to 2.6, 8.0, and 34.3 at 100 kPa, respectively, which suggests that it has the potential for the effective separation of CO from the gas mixtures. Table 2 lists the benchmark materials for CO adsorption. Cu(I) adsorbents have higher adsorption capacity than the conventional porous adsorbent. The Cu(I)-4/AC adsorbent prepared in this study has relatively high CO/CO_2_ selectivity among the selected adsorbents.

### 3.3. Isosteric Heat of Adsorption

Isosteric heat of adsorption is a significant thermodynamic parameter to characterize the interaction between the adsorbate and the adsorbent and to design a gas adsorption separation process, which can be calculated by Clausius–Clapeyron equation [41] as
$${\left[\frac{\partial lnP}{\partial \left(1/T\right)}\right]}_{q}=-\frac{\Delta H_{s}}{RT}$$
where P is the pressure, R is the ideal gas constant, T is the experimental temperature, q is the adsorption amount, and $\Delta H_{s}$ is the isosteric heat of adsorption. In this work, the experimental isotherms and the L-F model predicted isotherms of CO at different temperatures of 273 K, 293 K, and 298 K (as shown in Figure 6) were used to calculate $\Delta H_{s}$ of CO adsorption on AC and Cu(I)-4/AC. The conventional Langmuir–Freundlich (L-F) adsorption model correlated the experimental results and the fitting parameters, which are listed in Table 3. The experimental data fit well with L-F model, as can be seen by the high values of R^2^ (the coefficient of the experimental data and the fitting data). $\Delta H_{s}$ can be derived from the slopes of the plots of lnP versus 1/T at given adsorption amounts, as shown in Figure 7. It was shown that the isosteric heats of CO adsorption on Cu(I)-4/AC are remarkably much higher than those on AC. The result indicates that the π-complexation interaction between CO and Cu(I) is stronger than the van der Waals interaction of CO with the parent adsorbent. Usually, $\Delta H_{s}$ is <20 kJ·mol^−1^ for common physical adsorption and >80 kJ·mol^−1^ for chemical adsorption [42]. The values of $\Delta H_{s}$ on Cu(I)-4/AC maintained around 50 kJ·mol^−1^ in the whole pressure range, suggesting that the strength of complex adsorption is between physisorption and chemisorption. Such isosteric heat is not only propitious to adsorb CO, but also liable to desorb CO with a normal engineering operations (evidence as shown in Figure 9).

### 3.4. Adsorption Kinetics of CO

In addition, it is also crucial that the CO adsorption rate need to be quite rapid for potential applications of CuCl/AC adsorbent in the adsorption-driven separation of CO from gas mixtures containing CO, CO_2_, CH_4_, and N_2_. Typically, the requirement of the adsorption process in industrial applications is shorter than 1 min [43]. Here, we studied the time-dependent adsorption of CO on Cu(I)-4/AC adsorbent by releasing a small amount of CO and studying the adsorbed amount as a function of time as shown in Figure 8. Cu(I)-4/AC showed a relatively rapid adsorption rate, which reached 96% of the CO capacity within 25 s. The rapid CO adsorption rate suggests that Cu(I)-4/AC can meet the requirements for industrial application of adsorbent to separate CO from gas mixtures containing CO, CO_2_, CH_4_, and N_2_ in a PSA process.

### 3.5. Cycle Adsorption of CO on Cu(Ι)/AC

In the actual processes of gas separation, an ideal adsorbent not only needs to have high adsorption capacity and high selectivity but also needs to exhibit a stable cyclic adsorption performance in long-term adsorption/desorption cyclical operation. The pure CO cyclical adsorption/desorption isotherm at 298 K was evaluated for six times (The degassing between each cycle was carried out at 353 K under vacuum). As shown in Figure 9, the maximum amount of CO adsorption was reduced by about 1.3% after six cycles of adsorption and desorption, suggesting that the CO adsorption process using CuCl/AC adsorbent is stable under the investigated conditions. Its stable adsorption behavior indicates that the CuCl/AC has broad application prospects in selective adsorption of CO. It must be noted that the parent gases for the separation of CO must be pretreated to remove moisture in industrial processes, since the Cu(Ι) in the CuCl/AC can be oxidized to Cu(II) in the form of copper chloride once in contact with water vapor and O_2_ (especially under light condition [44,45]), which then cannot form complexation with CO.

## 4. Conclusions

CuCl/AC adsorbents for the separation of CO have been successfully obtained using CuCl_2_ and Cu(HCOO)_2_ as the initial material by a solid-state auto dispersion method. CuCl_2_ and Cu(HCOO)_2_ can be transformed into highly dispersed CuCl with activation at 533 K under vacuum atmosphere. The CO adsorption capacity increased with transformed CuCl loading until 4 mmol·g^−1^ and then decreased afterward. The CO adsorption capacity of Cu(I)-4/AC achieved 45.4 cm^3^·g^−1^, and the CO/CO_2_, CO/CH_4_, and CO/N_2_ selectivities were up to 2.6, 8.0, and 34.3 at 100 kPa, respectively. In addition, the isosteric heat of adsorption on Cu(I)-4/AC was about 50 kJ·mol^−1^. The CO adsorption capacity almost remains constant during six times cyclical adsorption and rapid adsorption kinetics at the adsorption process. Those excellent properties of Cu(I)-4/AC adsorbent would make it a promising adsorbent for CO separation and purification.

## Figures and Tables

**Figure 1 materials-12-01605-f001:**
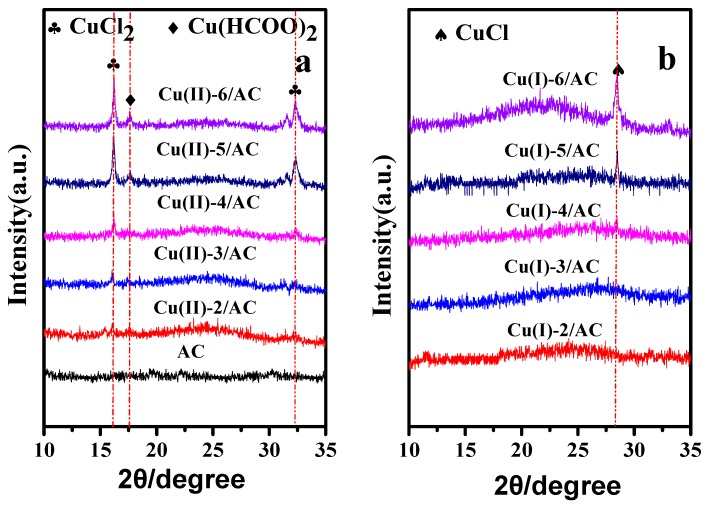
X-ray powder diffraction (XRD) patterns of activated carbon (AC), Cu (II)-x/AC (**a**) and Cu(I)-x/AC (**b**).

**Figure 2 materials-12-01605-f002:**
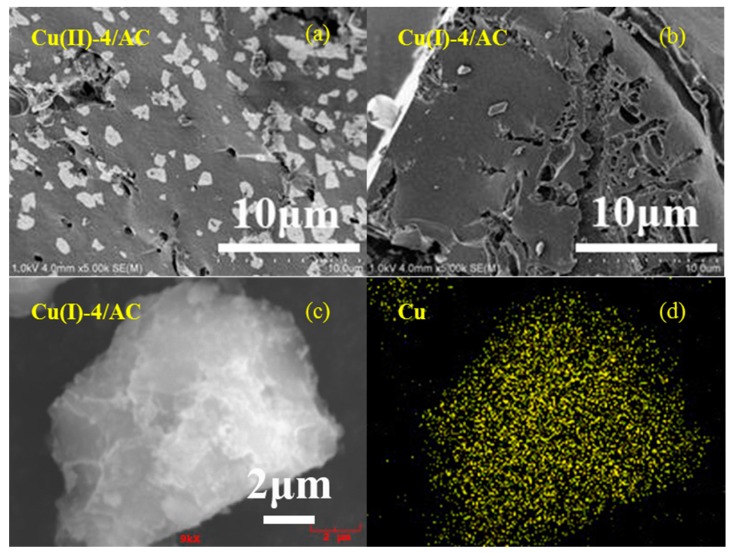
Scanning electron microscopy (SEM) images of Cu(II)-4/AC (**a**), Cu(I)-4/AC (**b**), and selected-area element mapping analyses of Cu(I)-4/AC (**c**,**d**).

**Figure 3 materials-12-01605-f003:**
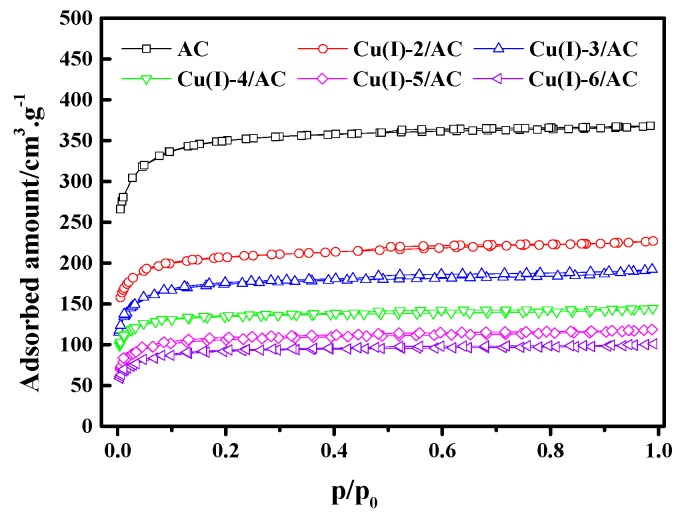
N_2_ adsorption/desorption isotherms of Cu(I)-x/AC.

**Figure 4 materials-12-01605-f004:**
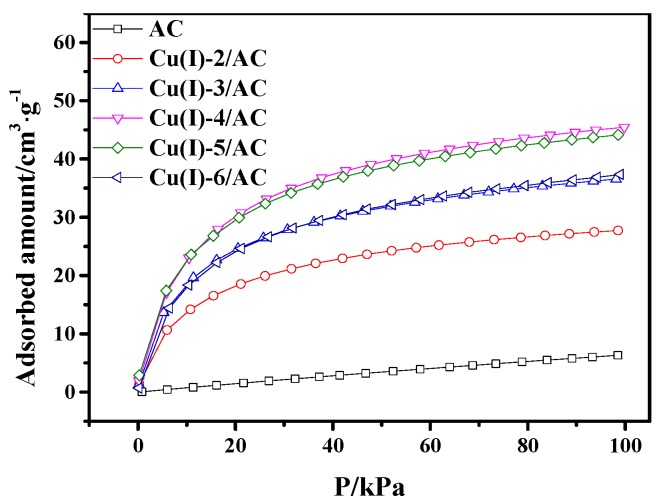
CO adsorption isotherms on AC and Cu(I)-x/AC at 298 K.

**Figure 5 materials-12-01605-f005:**
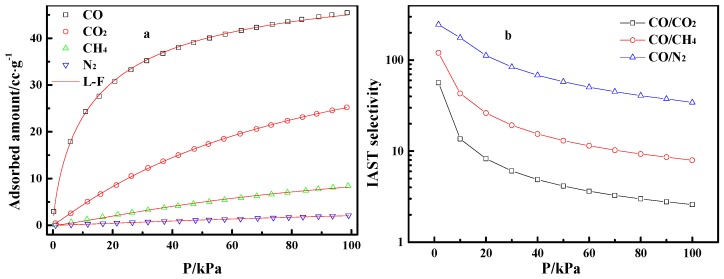
Adsorption isotherms of CO, CO_2_, CH_4_, and N_2_ on Cu(I)-4/AC and Langmuir–Freundlich (L-F) fitting lines (**a**), and ideal adsorbed solution theory (IAST)-predicted adsorption selectivities (**b**).

**Figure 6 materials-12-01605-f006:**
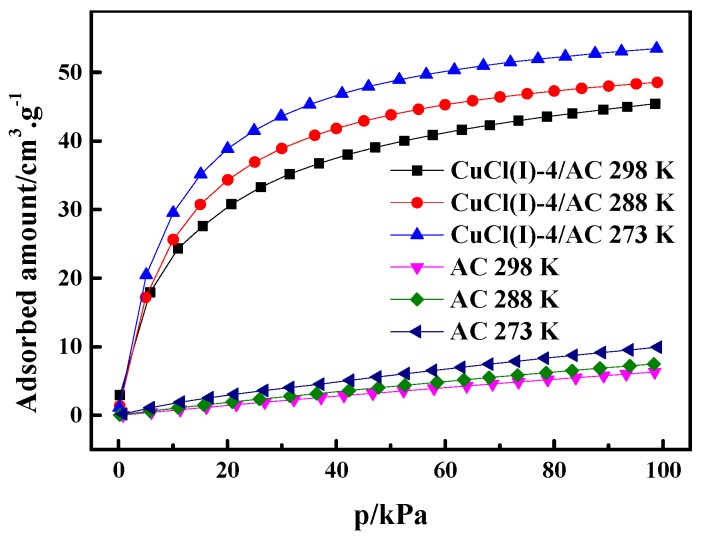
CO adsorption isotherms of AC and Cu(I)-4/AC at 273 K, 288 K, and 298 K.

**Figure 7 materials-12-01605-f007:**
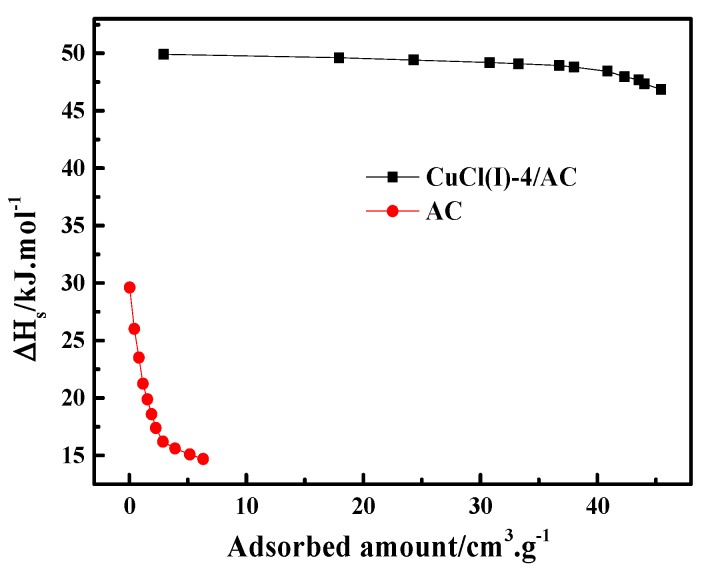
Isosteric heats of AC and Cu(I)-4/AC as the function of the adsorbed amount of CO.

**Figure 8 materials-12-01605-f008:**
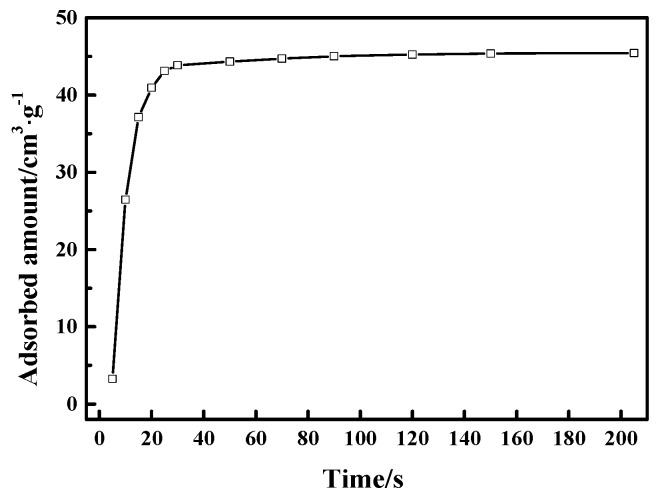
Uptake kinetics for Cu(I)-4/AC at 298 K.

**Figure 9 materials-12-01605-f009:**
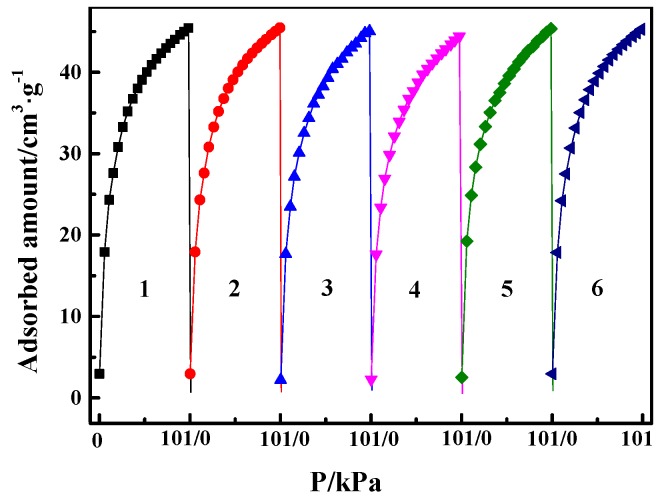
Adsorption and desorption cycles for CO at 298 K on Cu(I)-4/AC (the degassing between each cycle was carried out at 353 K under vacuum).

**Table 1 materials-12-01605-t001:** The parameters of pore structure, loading and utilization coefficient of CuCl for Cu(I)-x/activated carbon (AC).

Samples	S_BET_ (m^2^/g)	V_Total_ (cm^3^/g)	d^a^ (nm)	N-Cu^b^ (wt %)	Cu^c^ (wt %)	η (%)
AC	1082	0.571	2.11	/	/	/
Cu(I)-2/AC	804	0.372	1.85	10.7	10.2	72.5
Cu(I)-3/AC	644	0.294	1.83	14.8	14.1	69.0
Cu(I)-4/AC	505	0.223	1.76	18.3	17.5	68.9
Cu(I)-5/AC	395	0.182	1.84	21.3	20.0	58.0
Cu(I)-6/AC	331	0.156	1.89	24.0	22.5	43.6

^a^ average pore diameter of adsorbent; ^b^ calculated from starting material; ^c^ obtained by ICP-OES.

**Table 2 materials-12-01605-t002:** Comparisons with adsorbents in the literature.

Adsorbent	Refs.	T^a^(K)	$q_{100}^{b}\;({\mathrm{cm}}^{3}\cdot g^{-1})$				Selectivities		
			CO	CO_2_	CH_4_	N_2_	CO/CO_2_	CO/CH_4_	CO/N_2_
5A	[1]	298	26.9						
13x	[1,12]	298	13.5	102	8.9	5.0	0.1	1.5	2.7
BPL AC	[14]	298	4.4	24.4	8.5		0.2	0.5	
AC	[13]	303	11.5	58	24.8	7.4	0.2	0.5	1.6
CuCl(5)/Y	[22]	303	66.9	24.1	6.7	1.0	2.8	10.0	66.9
Cu(I)/AC	[32]	298	56.0	46.2	9.6	2.5	1.2	5.8	22.4
CuCl/NaY	[31]	303	52.0	29.3	3.9	1.8	1.7	13.3	28.8
CuCl/13X	[31]	303	84.9	53.1			1.6		
Cu(I)-4/AC	This work	298	45.4	25.2	8.4	2.1	2.6	3.0	34.3

^a^ the temperature of adsorption measurements; ^b^ adsorbed amount of gases at 100 kPa.

**Table 3 materials-12-01605-t003:** Langmuir–Freundlich (L-F) fitting parameters of CO isotherms on AC and Cu(I)-4/AC.

T(K)	AC				CuCl(I)-4/AC			
	*q* _m_	*b*	*n*	*R* ^2^	*q* _m_	*b*	*n*	*R* ^2^
298	10.22	1.82 × 10^−3^	0.688	0.994	51.87	0.0945	1.095	0.999
288	15.12	3.18 × 10^−3^	0.809	0.997	55.34	0.102	1.081	0.999
273	20.02	5.64 × 10^−3^	0.902	0.994	60.19	0.115	1.085	0.999

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
