# Peer review of "CO Adsorption Performance of CuCl/Activated Carbon by Simultaneous Reduction–Dispersion of Mixed Cu(II) Salts"

_materials, 2019, doi:10.3390/ma12101605_

Round 1
Reviewer 1 Report
The manuscript by Xue et al. described synthesis of CuCl/AC by simultaneous reduction-dispersion of CuCl2 and Cu(HCOO)2 and its CO adsorption properties. Although the experiments are systematically designed, the novelty of this work is relatively weak. The authors should highlight the novelty of this work in the introduction part. Also, there are so many typos and English errors. The authors should revise the manuscript thoroughly. The followings are some examples.
P. 2, line 60: K in N2 Xie et al. [31] …. (no period)
P. 2, line 71: Pure component … and CH4adsorption isotherms .. (no space)
P. 3, line 84: CuCl/AC adsorbents were synthetized .. (synthesized)
P. 3, line 98: The absorbed amount .. (adsorbed)
For analysis instruments, the description format should be the same including the country of manufacturing. (model, manufacturer, country). Line 96: (Quantachrome Inc., USA), Line 99: (SEM, Hitachi S4800), Line 101: (ICP-OES).
In Figure 2, it is recommended to include EDX dotmapping of Cu after activation to see the dispersion of Cu species.
Author Response
Reviewer 1:
The manuscript by Xue et al. described synthesis of CuCl/AC by simultaneous reduction-dispersion of CuCl2 and Cu(HCOO)2 and its CO adsorption properties. Although the experiments are systematically designed, the novelty of this work is relatively weak. The authors should highlight the novelty of this work in the introduction part. Also, there are so many typos and English errors. The authors should revise the manuscript thoroughly. The followings are some examples.
AUTHORS>> Thanks for the comments. In this work, we described a facile route of physically mixing CuCl2 and Cu(HCOO)2 powder with activated carbon (AC), followed by heating at 533 K under vacuum to make CuCl/AC adsorbent, which is new. It is important to have simple preparing steps in industry.
Some editing has been done carefully in the revised manuscript.<<< span="">
(1) P. 2, line 60: K in N2 Xie et al. [31] …. (no period)
AUTHORS>> Thank for the comments. We added period on P.2, line 60 in the revised manuscript.<<< span="">
(2) P. 2, line 71: Pure component … and CH4adsorption isotherms .. (no space)
AUTHORS>> Thanks for the comments. A space is added on P. 2, line 71, “CH4adsorption” was changed to “CH4 adsorption” in the revised manuscript. <<< span="">
(3) P. 3, line 84: CuCl/AC adsorbents were synthetized .. (synthesized)
AUTHORS>> Thanks for the comments. P. 3, line 84, “synthetized” was changed to “synthesized” in the revised manuscript. <<< span="">
(4) P. 3, line 98: The absorbed amount .. (adsorbed)
AUTHORS>> Thanks for the comments. P. 3, line 97, “absorbed” was changed to “adsorbed” in the revised manuscript. <<< span="">
(5) For analysis instruments, the description format should be the same including the country of manufacturing. (model, manufacturer, country). Line 96: (Quantachrome Inc., USA), Line 99: (SEM, Hitachi S4800), Line 101: (ICP-OES).
AUTHORS>> Thanks for the comments. We made the changes in the revised manuscript:
1) P. 3, line 95, “QUADRASORB SI, (Quantachrome Inc., USA)” was changed to “(QUADRASORB SI, Quantachrome Inc., USA)”.
2) P. 3, line 98, “(SEM, Hitachi S4800)” was changed to “(SEM, Hitachi S4800, Hitachi Ltd., Japan)”
3) P. 3, line 100 “(ICP-OES,)” was changed to “(ICP-OES, Thermo-ICAP6300, Seymer Feiser co., Ltd., USA)”
4) P. 3, line 104 “a static volumetric apparatus of the type NOVE1000e instrument (Quantachrome Inc., USA” was changed to “a static volumetric apparatus (NOVE1000e, Quantachrome Inc., USA).”<<< span="">
(6) In Figure 2, it is recommended to include EDX dotmapping of Cu after activation to see the dispersion of Cu species.
AUTHORS>> Thanks for the comments. We analyzed the EDX dot mapping, according to the reviewer’s comments. It was added in Figure 2 (c,d) in the revised manuscript.<<< span="">

Reviewer 2 Report
Herein, the authors present a CuCl2 modified activated carbon which shows exceptional CO adsorption performance and selectivity over a number of industrially relevant gases. The authors have performed an array of experiments to verify the performance and have determined CO selectivity using IAST. The material shows good CO uptake and stability during cycling experiments. I would recommend for publication in Materials pending the authors addressing the following:
· It would be good in the authors could provide a table of some of the benchmark materials for CO adsorption and if possible a list of their CO/CO2 selectivity’s.
· At what temperature was the desorption cycling carried out at in Figure 9? This should be included in the manuscript.
· Did the authors consider dynamic breakthrough experiments to further assess the CO selectivity of these materials under industrially relevant conditions? See Angew. Chem. Int. Ed., 2018, 57, 3332 –3336.
· It would be good if in future studies the authors considered the use of molecular modelling to determine the CO interactions with the CuCl2/AC material.
· The authors should address the “track changes” edits in the document and fix before publication (Page 4; line 127-128).
Author Response
Reviewer 2:
Herein, the authors present a CuCl2 modified activated carbon which shows exceptional CO adsorption performance and selectivity over a number of industrially relevant gases. The authors have performed an array of experiments to verify the performance and have determined CO selectivity using IAST. The material shows good CO uptake and stability during cycling experiments. I would recommend for publication in Materials pending the authors addressing the following:
(1) It would be good in the authors could provide a table of some of the benchmark materials for CO adsorption and if possible a list of their CO/CO2 selectivity’s:
AUTHORS>> Thanks for the comments. We prepared a table in the revised manuscript, comparing the adsorption of CO and CO2, CH4, N2 as well, selectivity of CO/CO2, CO/CH4, and CO/N2, with some materials in the literature. Please see Table 2.<<< span="">
(2) At what temperature was the desorption cycling carried out at in Figure 9? This should be included in the manuscript
AUTHORS>> Thanks for the comments. On P9, L237 in the revised manuscript, we added a sentence “The degassing between each cycle was carried out at 353 K” In the meantime, we changed the figure caption of Figure 9 into “Adsorption and desorption cycles for CO at 298 K on Cu(I)-4/AC (the degassing between each cycle was carried out at 353 K under vacuum)”.<<< span="">
(3) Did the authors consider dynamic breakthrough experiments to further assess the CO selectivity of these materials under industrially relevant conditions? See Angew. Chem. Int. Ed., 2018, 57, 3332 –3336.
AUTHORS>> Thanks for the comments. We agree that dynamic breakthrough experiments is an efficient way of analyzing gas adsorption, giving the information of selectivity under industrially relevant conditions. However, this paper focusing on studying a facile route for making CuCl/AC adsorption and its CO adsorption. We haven’t got the setup for such experiment, but already planned to build up such experiment in the future.<<< span="">
(4) It would be good if in future studies the authors considered the use of molecular modelling to determine the CO interactions with the CuCl2/AC material.
AUTHORS>> Thanks for the comments. It is a good suggestion. Molecular modeling will definitely help to understand the interaction of CO molecule with CuCl/AC. We will try that in the future focusing on understanding the mechanism ofπ-complexation based adsorption.<<< span="">
(5) The authors should address the “track changes” edits in the document and fix before publication (Page 4; line 127-128).
AUTHORS>> Thanks for the comments. We have already fixed the “track changes” in the revised manuscript.<<< span="">

Round 2
Reviewer 1 Report
The authors revised the manuscript well based on the reviewers' comments. I am in favor of publishing this manuscript in Materials.
Author Response
Thanks for the comments
This manuscript is a resubmission of an earlier submission. The following is a list of the peer review reports and author responses from that submission.